# Reverse *C*-glycosidase reaction provides *C*-nucleotide building blocks of xenobiotic nucleic acids

Martin Pfeiffer[1,2] & Bernd Nidetzky [1,2 ✉]

*C*-Analogues of the canonical *N*-nucleosides have considerable importance in medicinal chemistry and are promising building blocks of xenobiotic nucleic acids (XNA) in synthetic biology. Although well established for synthesis of *N*-nucleosides, biocatalytic methods are lacking in *C*-nucleoside synthetic chemistry. Here, we identify pseudouridine monophosphate *C*-glycosidase for selective 5-β-*C*-glycosylation of uracil and derivatives thereof from pentose 5-phosphate (D-ribose, 2-deoxy-D-ribose, D-arabinose, D-xylose) substrates. Substrate requirements of the enzymatic reaction are consistent with a Mannich-like addition between the pyrimidine nucleobase and the iminium intermediate of enzyme (Lys166) and open-chain pentose 5-phosphate. β-Elimination of the lysine and stereoselective ring closure give the product. We demonstrate phosphorylation-glycosylation cascade reactions for efficient, one-pot synthesis of *C*-nucleoside phosphates (yield: 33 – 94%) from unprotected sugar and nucleobase. We show incorporation of the enzymatically synthesized *C*-nucleotide triphosphates into nucleic acids by RNA polymerase. Collectively, these findings implement biocatalytic methodology for *C*-nucleotide synthesis which can facilitate XNA engineering for synthetic biology applications.

[1] Institute of Biotechnology and Biochemical Engineering, Graz University of Technology, NAWI Graz, Petersgasse 12, 8010 Graz, Austria. [2] Austrian Centre of Industrial Biotechnology (acib), Petersgasse 14, 8010 Graz, Austria. ✉email: bernd.nidetzky@tugraz.at

Most natural and synthetic nucleosides (e.g., uridine, U; Fig. 1a) have their sugar and nucleobase parts connected via a hemiaminal ether (*N*-glycosidic) linkage. Nucleoside analogs replacing the canonical *N*- by a rare-to-nature *C*-glycosidic linkage resist the common routes of biological degradation by *N*-glycoside hydrolases and phosphorylases[1–4]. Since in vivo stability is crucial for compound efficacy and applicability in therapy, synthetic *C*-nucleosides have considerable importance in medicinal chemistry. Nucleoside analogs generally represent promising lead structures in drug design for antiviral, antibacterial and antitumor activity[5–9]. Synthetic *C*-nucleosides, such as BCX4430 (Galidesivir) and GS-6620 (Supplementary Fig. 1), can be remarkably effective against filovirus infections and hepatitis C virus, respectively[7,10–13]. Originally developed for use against filoviruses, GS-5734 (Remdesivir; Supplementary Fig. 1) is now also evaluated against infections of the SARS-CoV-2 virus[10,13,14]. Other *C*-nucleosides, such as showdomycin, formycin, pseudouridimycin and minimycin (Supplementary Fig. 1), constitute an emerging class of microbial natural products that exhibit unusual structures and diverse biological activities, including antibiotic efficacy[15–21].

*C*-nucleosides are furthermore important as natural or synthetic building blocks of nucleic acids, RNA in particular. The U isomer pseudouridine (Ψ; Fig. 1b) occurs naturally in all types of RNA (tRNA, rRNA, snRNA, snoRNA, scaRNA, and mRNA)[22–26]. The Ψ formation is catalyzed by Ψ-synthase, which post-translationally isomerizes U to Ψ [27,28]. Owing to additional hydrogen bonding from the nucleobase N1 in Ψ compared to U (Fig. 1a, b), positional substitution of U with Ψ can increase RNA stability by locally enhancing the macromolecular rigidity[23]. Chemically synthesized Ψ and derivatives thereof are important building blocks for study of *C*-nucleoside biology as well as for numerous synthetic biology applications based on xenobiotic nucleic acids (XNA)[29–36]. For example, the uniform substitution of U by Ψ can render synthetic mRNA non-immunogenic and was shown to enhance the translation efficiency[30–32]. Additionally, Ψ-modified mRNA can boost the gene-cutting efficiency in CRISPR-based gene editing[37,38].

To advance *C*-nucleosides for their emerging applications in XNA engineering or in drug design, the target compounds must be made available through efficient and structurally precise synthetic methods. Chemical routes to *C*-nucleosides typically follow

**Fig. 1 Proposed reaction mechanism of YeiN and the enzymatic phosphorylation-glycosylation cascade used in this study.** The *N*-nucleoside uridine (U) (**a**) compared to the *C*-nucleoside pseudouridine (Ψ) (**b**) and reverse reaction of YeiN for synthesis of Ψ 5′-phosphate (ΨMP) from D-Rib5P (**1**) and Ura (**12**) (**c**). In panel **c**, B is the base residue from the enzyme. The key catalytic steps are the following: formation of a covalent iminium ion intermediate between the enzyme (Lys166) and the open-chain D-Rib5P, assisted by Glu31 (I); Mannich-like reaction for C-C coupling (II); β-elimination of the amine of Lys166 (III); cyclization through an oxa-Michael addition-like reaction (IV). For further discussion of the enzyme mechanism, see the text under the Results and the Discussion. **d** Phosphorylation-glycosylation cascades for one-pot synthesis of ΨMP and Ψ 5′-triphosphate (ΨTP) from unphosphorylated D-Rib are shown. Dephosphorylation of ΨMP yields Ψ. RbsK D-Rib-5-kinase, PK pyruvate kinase, CIP calf intestine phosphatase, NDK nucleoside diphosphate kinase. PEP phosphoenolpyruvate, PYR pyruvate.

the synthetic rationale of (1) reconstructing the heteroaryl nucleobase on a C1′-functionalized sugar or (2) coupling the prepared heteroaryl with a suitable glycosyl reagent, such as a hydroxy group-protected glycal or pentonolactone[5,39–43]. However, despite much progress from research over decades, a purely chemical approach to defined C-nucleosides remains challenging to establish[44–50]. A synthesis that exploits the high stereo- and regiocontrol of enzymatic reactions for installing the C-glycosidic linkage would be highly desirable. Natural enzymes of C-glycoside biosynthesis, such as tRNA pseudouridine synthase[27,28,51,52] and C-glycosynthase[15,18–21,53,54] have complex substrate requirements, leading to perceived limitations on their applicability. Therefore, unlike N-nucleosides for which N-nucleoside phosphorylases present excellent tools of glycoside synthesis[55–60], biocatalytic methods are lacking for C-nucleoside synthetic chemistry.

Here, we identify pseudouridine monophosphate (ΨMP) C-glycosidase for C-nucleoside synthesis. The enzyme's natural role is the recycling of Ψ via degradation of its 5′-phosphate into uracil (Ura(**12**)) and D-ribose 5-phosphate (D-Rib5P(**1**)), as shown in Supplementary Fig. 2[61,62]. Despite early evidence suggesting applicability[63–65], the reverse C-glycosidase reaction (Fig. 1c) is unexplored synthetically. Multiplicity of enzymes in C-nucleoside turnover and overlapped enzyme naming used previously complicate clear assignment of enzyme type to function. For this study, we use the biochemically and structurally well-characterized ΨMP C-glycosidase from E. coli (YeiN). We show that YeiN catalyzes the selective 5-β-C-glycosylation of Ura (**12**) and derivatives thereof from several pentose 5-phosphate substrates, including the 5-phosphate of D-ribose, 2-deoxy-D-ribose, 2-deoxy-2-fluoro-D-ribose, D-arabinose or D-xylose. The ΨMP and its corresponding sugar or nucleobase variants are obtained in a single-step, highly efficient transformation. Remarkably, although formally a reversion of hydrolysis, the overall enzymatic conversion benefits from the favorable thermodynamics of its central catalytic step, a carbon-carbon bond-forming (Mannich-like) addition. It thus proceeds with excellent synthetic yields (≥90%), even in water. Finally, we show incorporation of the enzymatically synthesized C-nucleotide triphosphates into RNA. Collectively, therefore, our study implements biocatalytic methodology for C-nucleotide synthesis, which can facilitate XNA engineering for synthetic biology applications and establish new routes to natural product C-glycosides.

## Results

**The substrate scope of YeiN in reverse C-glycosidase reaction.** Figure 2 shows the substrates used. Besides the native substrates D-Rib5P (**1**) and Ura (**12**), we tested a series of sugar phosphates (in combination with Ura, **12**) and nucleobase analogs (in combination with D-Rib5P, **1**) as alternative substrates (Fig. 2) of C-glycosylation by YeiN. Based on product formation detected with HPLC, 2-deoxy-D-Rib5P (D-dRib5P, **2**), D-xylose 5-phosphate (D-Xyl5P, **3**), D-arabinose 5-phosphate (D-Ara5P, **4**) and 2-deoxy-2-fluoro-D-ribose 5-phosphate (2F-D-Rib5PP, **5**) are active. L-Ribose 5-phosphate (L-Rib5P, **6**), D-glucose 6-phosphate (D-Glc6P, **7**), D-mannose 6-phosphate (D-Man6P, **8**), D-fructose 6-phosphate (D-Fru6P, **9**), D-ribulose 5-phosphate (D-Rul5P, **10**) and D-erythrose 4-phosphate (D-Ery4P, **11**) are inactive. We also considered the 5-phosphates of 2-amino-2-deoxy- and 3-amino-3-deoxy-D-ribose, L-arabinose, D-lyxose, and L-xylose (Supplementary Fig. 3), but enzymatic phosphorylation of these pentoses from ATP proved not possible. The nucleobase analogs used for reaction with YeiN feature substitutions at each Ura carbon (Fig. 2). 3-Methyl-Ura (**13**), 6-amino-Ura (**14**), 4-thio-Ura (**15**) and 2-thio-Ura (**16**) are active. Absence of activity with 1-methyl-Ura (**17**), 5-fluoro-Ura

(**18**) and 6-trifluoro-methyl-Ura (**19**) can be mechanistically relevant, as discussed below. Additionally, thymine (5-methyl-Ura, **20**), 6-aza-Ura (**21**), 6-Cl-Ura (**22**), cytosine (**23**), 2,4-thio-Ura (**24**), 5-ethynyl-Ura (**25**), maleimide (**26**), adenine (**27**), guanine (**28**), 4-pyrimidone (**29**) and pyrimidine (**30**) were evaluated but found to be inactive. Combinatorial assessment of all sugar 5-phosphates active with Ura (**12**) against the full panel of Ura analogs used with D-Rib5P (**1**) confirms the substrate scope of YeiN (Supplementary Table 1).

We obtained kinetic parameters for the substrates (Fig. 2 and Supplementary Table 2). Compared with D-Rib5P (**1**) and Ura (**12**), catalytic efficiencies ($k_{cat}/K_M$) for alternative substrates are decreased between ~10 to ~$10^3$-fold, reflecting substantial (≥5-fold) changes in both $k_{cat}$ (decrease) and $K_M$ (increase).

**Characterization of the YeiN reaction.** The substrate scope of YeiN is rationalized in terms of structural and mechanistic requirements of the enzymatic reaction. Molecular-docking results show that, with the terminal phosphate group accommodated in the enzyme binding pocket (Fig. 3a), a five-carbon chain is needed to precisely position the substrates carbonyl for iminium ion formation with the ε-amino group of Lys166. Docking of D-Rib5P (**1**) results in a potentially catalytic complex, with the ε-amino group of Lys166 positioned (3.3 Å) for nucleophilic attack at the C1 carbonyl carbon. Thr139 (2.6 Å) and Glu31 (3.2 Å) are placed suitably to stabilize negative charge developing on the C1 oxygen atom during carbinol intermediate formation (Fig. 3a, c). The carbinol intermediate may be stabilized by Thr131 and the main chain carbonyl oxygen of Gly132 (Fig. 3c). Glu31 is furthermore positioned to promote expulsion of water from the carbinol to give the iminium ion (protonated Schiff base). In terms of structural arrangement and proposed catalytic function, Thr139 and Glu31 represent a clear analogy to aldolase active sites[66–72]. In terms of sequence and three-dimensional structure, however, YeiN shows hardly any similarity with well-studied aldolases, such as deoxy-ribose phosphate aldolase[67,71], D-fructose bisphosphate aldolase[69], and 2-dehydro-3-deoxy-phosphogluconate aldolase[70,72]. Mechanistically, YeiN differs from these aldolases in that it requires the iminium ion as an electrophile for C–C coupling (Fig. 1c). The aldolases convert the iminium ion to an enamine (carbanion iminium by resonance) so that it can function as a nucleophile in their aldol reactions[66–70,72].

Contrary to D-Rib5P (**1**), the docking poses for D-Ery4P (**11**), D-Rul5P (**10**), and D-Fru6P (**9**) have Thr139 and Glu31 not properly aligned to the carbonyl oxygen, geometrically (D-Ery4P, **11**; Supplementary Fig. 4a) or in terms of interaction distance (D-Rul5P, **10**; D-Fru6P, **9**; Supplementary Fig. 4b, d, respectively). Additionally, the poses of D-Ery4P (**11**) and D-Rul5P (**10**) suggest loss of the stabilizing interaction with Gly132. The D-Glc6P (**7**) pose (Supplementary Fig. 4c) shows an enlarged distance between the ε-amino of Lys166 and the C1 carbon. Lack of activity of these sugar phosphates is thus plausibly explained.

Docking of Ura (**12**) into the structure of the covalent enzyme-D-Rib5P adduct (iminium ion form) places the reactive sugar C1′ and nucleobase C5 at a distance (3.4 Å) and orientation (91° angle between C5-C1′ and pyrimidine ring plane; Fig. 3b) well suited for carbon-carbon bond formation. Mechanistically, the catalytic reaction on the enzyme resembles a Mannich-like addition, for it involves the addition of a nucleophile (the cyclic enamine of the nucleobase) to an iminium ion (the activated aldehyde of D-Rib5P, **1**) (Figs. 1c and 3c).

Except for positioning, there seems to be limited catalytic facilitation from the enzyme to the agency of the Ura (**12**) C5 as a nucleophile of the reaction. The crystal structure of a covalent

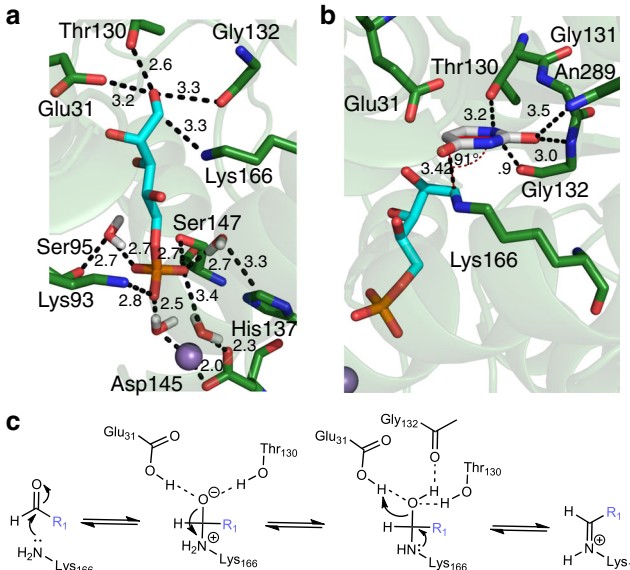

**Fig. 2 Substrate scope of YeiN.** Substrates tested for *C*-glycosylation by YeiN. The compounds framed in blue are active while those framed in magenta are not.

**a** Thr130, Gly132, Glu31, Lys166, Ser147, Ser95, Lys93, His137, Asp145

**b** Gly131, Thr130, An289, Glu31, Gly132, Lys166

**c** Glu31, Gly132, Thr130, Lys166

**Fig. 3 The substrate specificity of YeiN analyzed with molecular docking.** Docking poses of **a** D-Rib5P (**1**, cyan) and **b** Ura (**12**, white). The enzyme bound $Mn^{2+}$ is shown as a violet sphere. Distances are shown in Å. **c** Suggested mechanism of formation of the key covalent iminium ion intermediate that serves as electrophile in the subsequent Mannich-like addition, as shown in Fig. 1c.

covalently bound D-Rib5P (**1**). Further docking analysis summarized in Supplementary Fig. 6 shows that lack of reactivity with D-Rib5P (**1**) can reflect steric restraint on the productive binding of certain nucleobases, such as 6-trifluoromethyl-Ura (**19**), 6-Cl-Ura (**22**) and 1-methyl-Ura (**17**) (Supplementary Fig. 6a, b, d). Other Ura (**12**) analogs, including 5-fluoro-Ura (**18**) as well as 5-methyl-Ura (**20**) and 6-aza-Ura (**21**) (Supplementary Fig. 6e, f, c) are accommodated in the enzyme binding pocket almost identically as the parent Ura (**12**). Positioned properly for catalysis, these Ura (**12**) analogs seem to be rendered inactive intrinsically. The proposed path for the catalytic reaction of YeiN (Fig. 1c) suggests mechanistic rationale for why the particular substitutions are incompatible with the enzyme activity. The 1-methyl substituent precludes formation of the mechanistically important cyclic imine tautomer in the catalytic step of C–C coupling (Fig. 1c). Owing to electronic effect, the 6-aza-Ura (**21**) is an unlikely nucleophile for reaction with the iminium ion of the covalent enzyme intermediate. In the subsequent catalytic step, as shown in Fig. 1c, the Mannich adduct undergoes β-elimination of the amine of the lysine (Fig. 1c). Ura (**12**) analogs bearing substituents at the C5 are chemically incompetent to release the lysine analogously.

In the last step of the synthetic reaction, cyclization to a β-*C*-pentofuranosyl ring occurs and this proceed chemically through a kind of oxa-Michael addition (Fig. 1c). The cyclization could happen on the enzyme, in which case it is likely to be stereospecific, or in solution. Anomeric stereoselectivity can be a distinguishing feature of enzymes that catalyze *C*-ribosylation. The *C*-glycosynthase AlnA produces a mixture of α- (16%) and β-glycosidic (84%) forms during *C*-ribosylation of the naphthoquinone prealnumycin from D-Rib5P (**1**), suggesting ribosyl ring closure without enzymatic assistance (Supplementary Fig. 7)[54]. Here, we used in situ proton NMR in $^2H_2O$ solvent to monitor ΨMP formation from, and degradation to, D-Rib5P (**1**) and Ura

intermediate between YeiN and ΨMP shows just a single hydrogen bond between the C2 oxygen atom of Ura and Asn289 (Supplementary Fig. 5)[62]. The docking pose of Ura (**12**) suggests additional polar contacts (Fig. 3b) that can contribute to the precise positioning of the C5 of Ura above the C1 of the

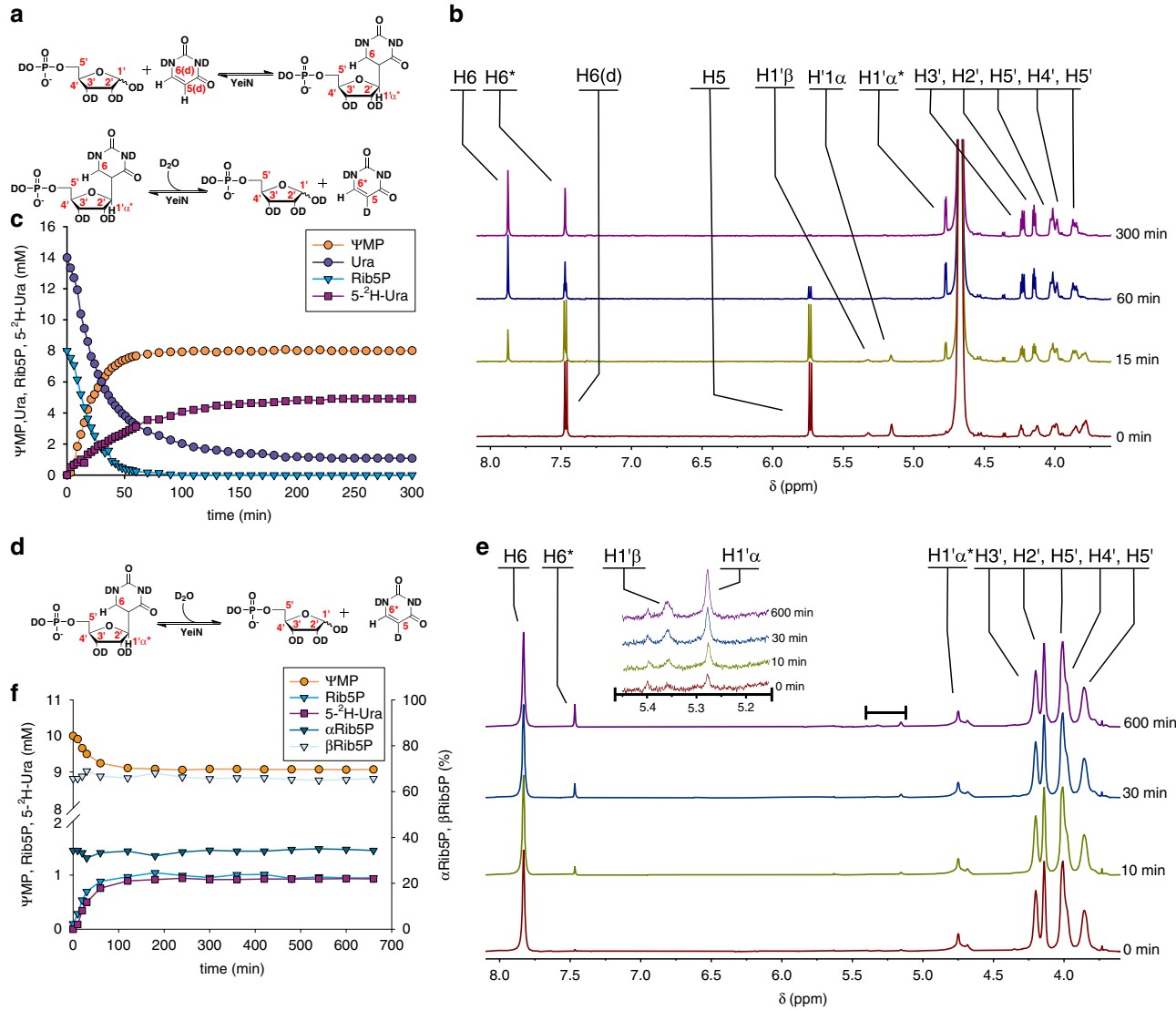

**Fig. 4 Enzymatic synthesis and hydrolysis of ΨMP monitored by ${}^{1}$H-NMR. a** Scheme of enzymatic synthesis of ΨMP. Solvent deuterium is incorporated into Ura (**12**) C5 due to enzymatic hydrolysis of ΨMP. **b** ${}^{1}$H-NMR traces of ΨMP synthesis from 14 mM Ura (**12**) and 8 mM D-Rib5P (**1**), using 3 μM YeiN. **c** Time course of ΨMP synthesis as determined from the NMR spectra ($n = 1$). **d** Scheme of solvent deuterium incorporation into Ura C5 due to the enzymatic hydrolysis of ΨMP **e** ${}^{1}$H-NMR traces of hydrolysis of Ψ (10 mM) by 3 μM YeiN. **f** Time course of ΨMP hydrolysis as determined from the NMR spectra ($n = 1$). Relative amounts of β-D-Rib5P and α-D-Rib5P are displayed on the secondary axis.

(**12**). The time-resolved NMR spectra for the synthesis direction (Fig. 4a–c; 14 mM Ura, **12** and 8 mM D-Rib5P, **1**) show that only a single anomer (β-C-riboside) is formed. They also show that solvent deuterium is incorporated at the Ura (**12**) C5 at a rate (0.07 mM/min) about 3-fold lower than the rate of conversion of Ura (**12**) into ΨMP (0.22 mM/min). This suggests that the deuterium uptake does not happen immediately during ΨMP synthesis but occurs only later when ΨMP and Ura (**12**) equilibrate dynamically. Suitable controls show that deuterium uptake into Ura (**12**) necessitates ΨMP formation (Supplementary Fig. 8). Inactive sugar phosphates (D-Ery4P, **11**; D-Rul5P, **10**; D-Glc6P, **7**; and D-Fru6P, **9**) do not promote deuterium uptake into the C5 of Ura (**12**). Analyzing the hydrolysis of ΨMP (Fig. 4d–f), we confirm that the Ura (**12**) released from ΨMP incorporates solvent deuterium at its C5. The hydrolysis reaction approaches equilibrium after about 100 min and an equilibrium constant of $8.9 \times 10^{-5}$ M can be calculated from the data (0.9 mM Ura, **12** and D-Rib5P, **1**; 9.1 mM ΨMP).

The NMR analysis (Fig. 4e, f) additionally shows that at all times, the D-Rib5P (**1**) released from ΨMP involves α-furanose (33%) and β-furanose (67%) forms at anomeric equilibrium. Plausible interpretation is that the open-chain form of D-Rib5P is released into solution where the furanose structures are formed spontaneously. However, the mutarotation of D-Rib5P (**1**)[73] occurs at the limits of temporal resolution of the experiment, leaving the conformation of the released D-Rib5P (**1**) undecided.

**Phosphorylation-glycosylation cascade for the synthesis of C-nucleoside monophosphates**. We considered that limitation on the synthetic utility of the YeiN-catalyzed formation of C-nucleoside monophosphates can arise from the requirement for a pentose 5-phosphate as the substrate. The unphosphorylated pentose, however, is an attractive starting material for synthesis. We therefore designed an enzymatic cascade reaction (Fig. 1d) in which pentose phosphorylation[74] and C-glycosylation of the

nucleobase are telescoped into a one-pot transformation. We show efficient enzymatic synthesis of ΨMP from D-Rib and Ura (12) and demonstrate applicability of the biocatalytic system to the synthesis of all sugar and nucleobase variants of the C-glycoside that were shown in Fig. 2 to be accessible via the YeiN reaction. As shown in Supplementary Fig. 9 and summarized in Table 1, the yield of synthesis ranges between useful and excellent (33–95%), and product (10–60 mg) can be recovered generally with high purity from small-scale reactions (15 mL). Product structures are confirmed by NMR (Supplementary Fig. 10–29) and agree well with published spectra[75].

**Dephosphorylation and phosphorylation for structural diversification of the C-nucleoside triphosphates**. We considered that chemical biology and medicinal chemistry applications can require the unphosphorylated C-nucleoside or the C-nucleoside triphosphate. We therefore show enzymatic dephosphorylation of ΨMP with non-specific phosphatase to obtain the C-nucleoside in excellent yield (Table 1 and Supplementary Fig. 30). Dephosphorylation can be analogously applied to the other C-glycoside monophosphates. To prepare the C-nucleoside triphosphate (e.g., ΨTP), we set up a cascade reaction in which the monophosphate is phosphorylated by nucleotide monophosphate kinase and pyruvate kinase in the presence of ATP and PEP (Fig. 1d). Interestingly, the addition of a nucleotide diphosphate kinase did not improve efficiency of the overall phosphorylation (Supplementary Fig. 31)[76]. We show synthesis of ΨTP, 2-deoxy-ΨTP (dΨTP) and 6-amino ΨTP in a one-pot reaction starting from unphosphorylated sugar and Ura (12) (Fig. 5 and Supplementary Fig. 32). Products were recovered in mg amounts and in suitable purity (≥85%) for characterization and further studies.

**C-Nucleoside triphosphates for enzymatic synthesis of xenobiotic RNA**. Using the YeiN-encoding gene as the template, we examined amplification of the corresponding RNA and DNA using ΨTP, 6-amino-ΨTP and dΨTP in place of, or together with, the native UTP and dTTP, respectively. Applying T7 RNA polymerase as shown in Fig. 6a, we succeeded in synthesizing RNA built solely from Ψ instead of U.

Densitometric analysis of gel bands suggested that RNA polymerase transcription efficiency (expressed as gel band intensity) was similar when ΨTP was used to substitute UTP. Unexpectedly, the band intensity decreased to about half when ΨTP and UTP were added at the same time. Overloading of the RNA polymerase with NTP substrate may have caused inhibition, perhaps due to limitation in the concentration of Mg$^{2+}$ used. 6-Amino-ΨTP and dΨTP were not accepted by the used RNA polymerase. Applying DreamTaq-polymerase that is known for its relaxed dNTP substrate specificity, we observed that dΨTP cannot substitute for dTTP (Fig. 6b). However, when used together with dTTP, the dΨTP appeared to inhibit the amplification slightly (~25%).

## Discussion

We here demonstrate the reverse reaction of ΨMP C-glycosidase for efficient biocatalytic synthesis of C-nucleoside monophosphates. Besides the natural ΨMP, a panel of sugar and nucleobase analogs of the C-nucleoside monophosphate are synthesized with excellent regiocontrol and stereoselectivity. The substrate requirements of the enzymatic reaction are consistent with the mechanistic proposal (Fig. 1c) of a Mannich-like C-C addition between the iminium ion of an enzyme bound open-chain pentose 5-phosphate intermediate and the pyrimidine nucleobase. β-Elimination of the enzyme residue and stereospecific cyclization via oxa-Michael-like addition give the C-nucleoside

monophosphate product. Use of the iminium ion intermediate as an electrophile of the reaction distinguishes YeiN from well-studied aldolases that form essentially the same covalent intermediate but convert it to an enamine nucleophile for aldol coupling.

Chemical studies of C-glycoside synthesis have shown that the condensation product is favored thermodynamically, even in water[77,78], and the synthetic use of the chemical reaction was demonstrated[73–77]. Equilibrium on the YeiN reaction far on the side of the C-nucleoside monophosphate, which would seem unusual for a nominal glycosidase reaction, is thus explained. Enzymatic C–C coupling reactions are synthetically important in the carbohydrate sciences[65,78–86] and the current study adds new application-relevant feature to them.

The development of enzymatic phosphorylation-glycosylation cascade reactions enable expedient, one-pot biocatalytic syntheses of the C-nucleoside 5′-phosphates from unphosphorylated pentoses as the substrates (Fig. 1d). We show that the C-nucleoside 5′-phosphates are readily dephosphorylated to give the C-nucleoside and that they can also be converted into the corresponding 5′-triphosphates. The non-phosphorylated C-nucleoside and its 5′-triphosphate are promising substrates for chemical and polymerase-catalyzed XNA synthesis, respectively. Phosphorylation-glycosylation cascades extend the synthetic utility of the ΨMP C-glycosidase reaction, enabling C-nucleoside phosphates to be obtained from unprotected sugar and nucleobase.

The ΨMP C-glycosidases are related evolutionary to a class of C-glycosynthases, represented by the well-characterized AlnA from *Streptomyces sp.* CM020, that catalyze β-C-riboside formation from D-Rib5P (1) and different naphthoquinones (Supplementary Fig. 7)[53,54,87]. The herein developed idea of modular phosphorylation-glycosylation cascades could be analogously applied to AlnA-type C-glycosynthases, to also harness these enzymes' catalytic proficiency and substrate promiscuity for practical synthesis.

## Methods

**Materials**. ATP ( > 95% purity) was from Roth (Karlsruhe, Germany). Deuterium oxide (99.96%) was from Euriso-Top (Saint-Aubin Cedex, France). Calf intestine phosphatase (CIP) was from New England Biolabs (Frankfurt am Main, Germany). Rabbit muscle pyruvate kinase (PK), adenosine triphosphate (ATP), D-ribose (D-Rib), D-2-deoxy-ribose (D-dRib), D-xylose (D-Xyl), D-arabinose (D-Ara), D-mannose (D-Man), D-glucosamine (D-GlcN), N-acetyl-D-glucosamine (D-GlcNAc), D-galactose (D-Gal), D-galactosamine (D-GalN), D-lyxose (D-Lyx), L-arabinose (L-Ara), L-xylose (L-Xyl), D-ribose-5-phosphate (D-Rib5P(**1**)), D-fructose 6-phosphate (D-Fru6P(**9**)), D-mannose 6-phosphate (D-Man6P (**8**)), D-glucose 6-phosphate (D-Glc6-P (**7**)), D-erythrose 4-phosphate (D-Ery4P (**11**)), D-ribulose 5-phosphate (D-Rul5P (**10**)), uracil (Ura (**12**)), 6-amino-Ura (**14**), 4-thio-Ura (**15**), 2-thio-Ura (**16**), 6-trifluoromethyl-Ura (**19**), 1-methyl-uracil (**17**), pyrimidine (**29**), pyrimidone (**30**), maleimide (**26**), adenine (**27**), guanin (**28**), thymine (**20**), and cytosine (**23**) were from Sigma Aldrich (Vienna, Austria). 3-Methyl-Ura (**13**), 6-aza-Ura (**21**), 6-Cl-Ura (**22**), and L-ribose 5-phosphate (**6**) were from Carbosynth (Compton, UK). 2-Amino-2-deoxy-D-ribose (2A-D-Rib), 3-amino-3-deoxy-D-ribose (3A-D-Rib), and 2-deoxy-2-fluoro-D-ribose (2F-D-Rib)) were from Synthose (Concord, Ontario, Canada). Expression vectors (pet15b) containing the genes for ribokinase (RbsK) or β-pseudouridine 5′-phosphate glycosidase (YeiN) were obtained from Genescript (Leiden, The Netherlands). Restriction enzymes, Phusion polymerase, T7 RNA Polymerase, Dream Taq DNA polymerase, and GenJET® PCR clean-up kit were from Thermo Fisher Scientific (Altham, MA, USA). DNA sequencing was done at LGC Genomics (Berlin, Germany).

**Ψ-5′-phosphate glycosidase (YeiN)**. An N-terminally His$_6$-tagged E. coli YeiN was used. The expression vector (pet15_yein) as obtained from Genescript was transformed into the E. coli BL21_DE3 (pLys) expression strain by electroporation and transformants were selected on 0.1 mg/mL ampicillin LB-agar plates.

YeiN was expressed in 1-L baffled shaken flasks at 37 °C and 110 rpm using 250 mL Lennox-media containing 0.1 mg/mL of ampicillin. Inoculation was to an initial OD$_{600}$ of 0.1. At an OD$_{600}$ of 0.8, the temperature was decreased to 18 °C and gene expression was induced with 0.4 mM IPTG (isopropyl β-D-1-thiogalactopyranoside) for 20 h. Cells were harvested by centrifugation at 2000 × g

**Table 1 Enzymatic syntheses with yields and product purities indicated.**

| Product | Pentose 5-P yield[#] [%] | Product yield[§] (%) | Isolated product (mg) | Purity (%) | Product | Pentose 5-P yield[#] (%) | Product yield[§] (%) | Isolated product (mg) | Purity (%) |
|---|---|---|---|---|---|---|---|---|---|
| ΨMP | 99 | 91 | 59 | 99 | ΨTP | 99 | 85 | 28 | 85 |
| dΨMP | 91 | 88 | 38 | 75 | dΨTP* | 91 | 60 | 15 | 98* |
| 2′-deoxy-2′-fluoro-ΨMP[i] | 90 | 85 | 5 | 70 | 6-amino-ΨTP* | 99 | 4 | 4 | 80 |
| D-Ara-ΨMP[ii] | 31 | 55 | 16 | 66 | 2-thio-ΨTP | 99 | n.d. | n.i. | |
| D-Xyl-ΨMP[iii] * | 56 | 33 | 8 | 95* | 4-thio-ΨTP | 99 | n.d. | n.i. | |
| 6-amino-ΨMP | 99 | 85 | 35 | 70 | 3-methyl-ΨTP | 99 | n.d. | n.i. | |
| 2-thio-ΨMP | 99 | 50 | 36 | 99 | D-Ara-ΨTP* | 31 | 1 | n.i. | > 40 |
| 4-thio-ΨMP | 99 | 94 | 36 | 99 | D-Xyl-ΨTP | 56 | n.d | n.i. | |
| 3-methyl-ΨMP | 99 | 94 | 10 | 70 | Ψ | 99 | 87 | 42 | > 99 |

*n.d.* no product formation detected.
*n.i.* not isolated.
*NMR spectra not recorded. Purity based on HPLC.
[§]Based on Ura (**12**) converted.
[#]Based on pentose converted.
[i]5-(2-deoxy-2-fluoro-β-D-ribofuranosyl 5-phosphate)uracil, [ii]5-(β-D-arabinofuranosyl 5-phosphate)uracil, [iii]5-(β-D-xylofuranosyl 5-phosphate)uracil.

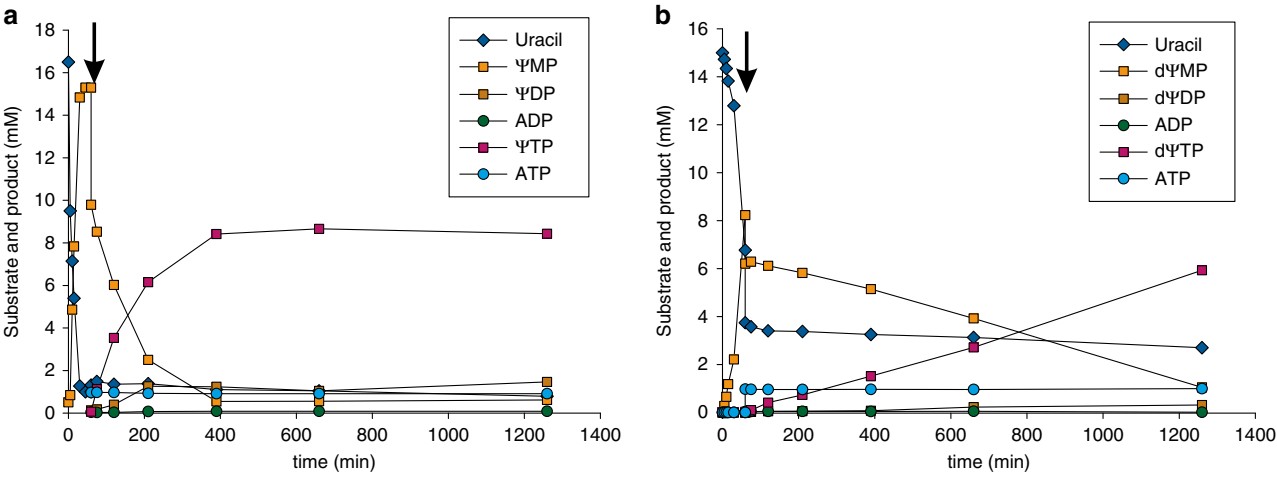

**Fig. 5 Enzymatic cascade synthesis of ΨTP and dΨTP.** Panels **a** and **b** show ΨTP and dΨTP, respectively. Pentoses (100 mM) were phosphorylated applying RbsK (15 μM for D-Rib; 30 μM for D-dRib) and pyruvate kinase (4 μM) using 100 mM PEP and 2 mM ATP as phosphate donors (see Supplementary Fig. 33). ΨMP synthesis was initiated by addition of 15 mM Ura, 1 mM MnCl₂, YeiN (1 μM for D-Rib, 3 μM for D-dRib) and 10 mM HEPES buffer, resulting in a concentration of 25 mM pentose 5-phosphate (time point zero **a**, **b**. The arrow indicates the start of the phosphorylation reaction by addition of 15 mM PEP, 15 μM CMPK, and 1 mM ATP, resulting in a volume increase of 33% ($n = 1$).

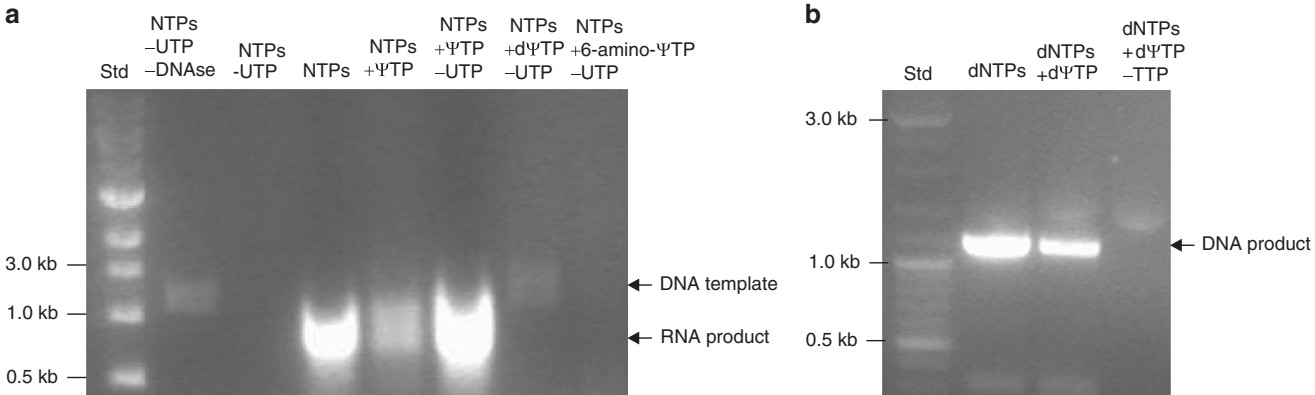

**Fig. 6 Polymerase-catalyzed amplification of RNA and DNA.** Panels **a** and **b** show RNA and DNA, respectively. RNA was transcribed from YeiN-encoding template DNA ($n = 1$). **a** In vitro translation was performed using standard conditions, 30 U of T7 RNA polymerase, 700 ng template DNA, transcription buffer (40 mM Tris-HCl, pH 7.9, at 25 °C; 6 mM MgCl₂, 10 mM DTT, 10 mM NaCl and 2 mM spermidine) and 2 mM of each NTP (ATP, GTP, UTP, CTP). Modified RNA was synthesized either by addition of 2 mM of ΨTP to the NTPs (NTPs + ΨTP) or by fully replacing UTP by 2 mM ΨTP (NTPs + ΨTP − UTP). **b** The *yein* gene was amplified from 50 ng expression vector (pet15_yein) using 1.25 U of Dream-Taq DNA polymerase, 0.2 mM of each dNTP (dATP, dCTP, dGTP, dTTP) and 0.5 μM of forward and reverse primer. To incorporate dΨ into the DNA either 0.2 mM of ΨTP were additionally added to dNTP mix (dNTPs + dΨTP) or TTP was fully replaced by dΨTP (dNTPs + YTP − TTP). Further details are described in "Methods".

at 4 °C for 30 min using a Sorvall RC-5B refrigerated superspeed centrifuge (Du Pont Instruments, Newtown, CT, USA). The supernatant was discarded, and the pellet was suspended in 50 4-(2-hydroxyethyl) piperazine-1-ethanesulfonic acid (HEPES) buffer (pH 8.0). Cells were disrupted using sonication (Fisherbrand Sonic Dismembrator, model Ultrasonic Processor FB-505; Fisher Scientific, Vienna, Austria; 5 min) on ice and the cell-free supernatant was recovered by centrifugation at 4 °C and 2000 × *g* for 30 min. Pre-treated cell lysate (20 mL) was loaded on 2 × 5 mL HisTrap FF column (GE Healthcare, Little Chalfont, UK), equilibrated with buffer A (50 mM HEPES, pH 8.0 containing 125 mM NaCl, 20 mM imidazole) and mounted on an ÄKTA prime plus (GE Healthcare) system. Protein was eluted using an imidazole gradient from 0% to 80% buffer B (50 mM HEPES, pH 8.0, containing 125 mM NaCl and 500 mM imidazole). The temperature was 10 °C and a flow rate of 3 mL/min was used. Fractions containing the target protein were pooled, concentrated and buffer-exchanged with Amicon Ultra-15 Centrifugal Filter Units (Millipore; Billerica, MA, USA). The final protein concentration was 30 mg/mL in 50 mM HEPES buffer containing 2 mM MgCl₂ (pH 7.0). Enzyme was stored at −20 °C until further use. Protein purification was monitored by sodium dodecyl sulphate–polyacrylamide gel electrophoresis (SDS PAGE) (Supplementary Fig. 34).

The YeiN activity assay was performed at 37 °C in 50 mM HEPES buffer supplemented with 2 mM MnCl₂ (pH 7.0). About 1.5 μM YeiN was used in the reaction. The substrates used were D-Rib5P (**1**) (2 mM) and Ura (**12**) (1 mM). One unit of YeiN activity is the amount of enzyme producing 1 μmol of ΨMP/min under the conditions used. Enzyme preparations had typically a specific activity of ~3 U/mg.

**Ribokinase (RbsK).** The expression vector (pet15_RbsK) as obtained from Genescript was transformed into the *E. coli BL21_DE3 (pLys)* expression strain by electroporation and transformants were selected on 0.1 mg/mL ampicillin LB-agar plates. RbsK was expressed and purified as described for YeiN, except the protein was stored in 50 mM HEPES supplemented with 2 mM MgCl₂ (pH 7.0).

An N-terminally His₆-tagged *E. coli* RbsK was used. The RbsK activity assay was performed at 37 °C in 50 mM HEPES buffer (pH 7.0, pH adjusted with KOH) containing 2 mM MgCl₂ and 3 μM RbsK. The substrates used were ATP (10 mM) and D-Rib (10 mM). One unit of RbsK activity is the amount of enzyme producing 1 μmol of D-Rib5P (**1**)/min under the conditions used. The enzyme preparation had a specific activity of ~50 U/mg.

**Pentose-5-phosphate phosphatase BT4131.** The sugar phosphate phosphatase BT4131 from *Bacteroides thetaiotaomicron* VPI-5482 was used as an N-terminal fusion with the positively charged binding module Z_basic2[88]. Expression, harvest and cell disruption was performed as described for YeiN except 50 μg/mL kanamycin was used instead of ampicillin.

Cell-free extract (20 mL) was loaded on 2 × 5 mL HiTrap SP FF column (GE Healthcare), equilibrated with buffer A (50 mM MES buffer, pH 7.0, containing 100 mM NaCl) and mounted on an ÄKTA prime plus (GE Healthcare) system. Protein was eluted using an NaCl gradient from 0% to 100% buffer B (50 mM HEPES buffer supplemented with 2 M NaCl, pH 7.0) at 10 °C using a flow rate of

3 mL/min. Fractions containing the target protein were pooled, concentrated and buffer-exchanged with Amicon Ultra-15 Centrifugal Filter Units (Millipore; Billerica, MA, USA). The final protein concentration was 10 mg/mL in 50 mM HEPES containing 2 mM $MgCl_2$ (pH 7.0). The enzyme was stored at 4 °C until further use. Protein purification was monitored using SDS PAGE.

The specific activity was determined in 50 mM HEPES buffer supplemented with 5 mM $MgCl_2$ (pH 7.0) at 37 °C using 20 mM D-Glc6P as the substrate. The reaction was started with 1.5 μM BT4131 and samples taken were quenched by heat treatment. Inorganic phosphate was measured at 850 nm[88]. One unit of BT4131 activity is the enzyme amount releasing 1 μmol of phosphate from D-Glc6P/min under the conditions applied. The specific activity of the phosphatase preparation was typically ~16 U/mg.

**Nucleoside monophosphate kinases (NPK).** The uridine monophosphate kinase (umpk) and cytidine monophosphate kinase (cmpk) genes were amplified from genomic *E. coli* BL21_DE3 (pLys) DNA by PCR using Phusion polymerase and the following set of primers (restriction sites underlined): UMPK_NdeI_fw (CATATG GCTACCAATGCAAAACCCGTC), UMPK_XhoI_rv (CTCGAGTTATTCCGTGA TTAAAGTCCCTTCTTTTTC); CMPK_NdeI_fw (CATATGACGGCAATTGCCC CGGTTATTACC), CMPK_XhoI_rv (CTCGAGTTATGCGAGAGCCAATTTCTGG CGC). Doubly digested (NdeI and XhoI) PCR products were purified from agarose gels treated with shrimp alkaline phosphatase (NEB, Ipswich, MA, USA) and ligated (T4 ligase) into the linearized (NdeI and XhoI) pET-STREP3 expression vector. The constructed vector was transformed into electro-competent *E. coli* BL21_DE3 (pLys) and transformants were selected on LB-agar plates (0.05 mg/mL kanamycin). Isolated vectors were sequenced (LGC Genomics). NPK expression, cell harvesting and cell disruption was performed as described for YeiN, except that 0.05 mg/mL kanamycin instead of ampicillin was used and cells were dissolved in wash buffer (100 mM Tris-Cl pH 8.0, 150 mM NaCl, 1 mM EDTA). Cell-free extract (20 mL) was loaded (2 mL/min) on 1 × 5 mL StrepTrap HP column (GE Healthcare), equilibrated with wash buffer and mounted on an ÄKTA prime plus (GE Healthcare) system. Unspecifically bound protein was washed off with 10 column volumes of wash buffer at a flow of 3 mL/min. Protein was eluted using elution buffer (100 mM Tris-Cl pH 8.0, 150 mM NaCl, 1 mM EDTA, 2.5 mM desthiobiotin). Fractions containing the target protein were pooled, concentrated and buffer-exchanged with Amicon Ultra-15 Centrifugal Filter Units (Millipore). The final protein concentration was 15 mg/mL in 50 mM HEPES containing 2 mM $MgCl_2$ (pH 7.0). Enzyme was stored at −20 °C until further use. Protein purification was monitored using SDS PAGE (Supplementary Fig. 34). The UMPK was only weakly expressed and most of the recombinant enzyme was found in inclusion bodies.

An *N*-terminally strep-tagged CMPK was used. The NPK-assay was performed at 37 °C in 50 mM HEPES buffer (pH 7.0) containing 1 mM $MgCl_2$ and 5 μM CMPK. The substrates used were 3 mM UMP and 3 mM ATP. One unit of NPK activity is the amount of enzyme producing 1 μmol of UDP/min under the conditions used. CMPK preparation typically exhibited a specific activity of 20 U/mg.

**Nucleoside diphosphate kinase (NDK).** An *N*-terminally Strep-tagged variant was used. The ndk gene was amplified from genomic *E. coli* BL21_DE3 (pLys) DNA by PCR using Phusion polymerase and the following set of primers (overlapping 5′-overhangs in lower case letters): NDK_fwd (ATGGCTATTGAACGTACTTTT TCCATC), NDK_rv (TTAACGGGTGCGCGGGC). The PCR product was cloned into the pET-STREP3 expression vector by circular polymerase extension cloning. Briefly, using the primer pairs CEPEC_NDK_fwd (GTTCGAGAAAGGCTTAAT TAACCATATGgctattgaacgtactttttccatc), CPEC-NDK_rv (GTGGTGGTGGTGGT GCTCGAGttaacgggtgcgcgggc) and CPEC-NDK-Strep_fwd (ggaaaaagtacgttcaatagc CATATGGTTAATTAAGGCCTTTCTCGAAC) CPEC-NDK-Strep_rv (gcccgcgcac ccgttaaCTCGAGCACCACCACCACCAC) in two independent PCR reactions, complementary overlaps are introduced into the ndk gene and into the target vector.

The purified PCR products were used in an overlap-extension-PCR to prime each other and to create the final expression vector, which was transformed into *E. coli* XL1 Blue. Transformants were selected on LB-agar plates containing 0.1 mg/mL of ampicillin, and the isolated plasmid was sequenced. The sequenced vector was transformed into the expression strain *E. coli* BL21_DE3 (pLys) and transformants were selected on LB-agar plates containing 0.1 mg/mL of ampicillin. NDK expression and purification were as described for NDKs (Supplementary Fig. 34).

The NDK activity assay was performed at 37 °C in 50 mM HEPES buffer (pH 7.0) containing 1 mM $MgCl_2$ and 5 μM NDK. The substrates used were 3 mM uridine diphosphate (UDP) and 3 mM ATP. One unit of NDK activity is the amount of enzyme producing 1 μmol of UTP/min under the conditions used. The enzyme preparation had a specific activity of 21 U/mg.

**Determination of protein concentration.** Protein concentrations were measured using DS-11 Spectrophotometer (DeNovix; Wilmington, DE, USA) at 280 nm. Protein concentration was calculated using the corresponding molar extinction coefficient and the molecular weight computed by protparam (https://web.expasy. org/protparam/). RbsK (12,490 $M^{-1}cm^{-1}$, 32,290 Da), YeiN (10,095 $M^{-1}cm^{-1}$, 32,909 Da), CMPK (13,075 $M^{-1}cm^{-1}$, 25,970 Da), NDK (4470 $M^{-1}cm^{-1}$, 15,463 Da), BT4131 (14,690 $M^{-1}cm^{-1}$, 28,856 Da).

**Determination of nucleotide concentration.** Uracil content was measured photometrically using a DS-11 Spectrophotometer (DeNovix). Ura (**12**) (260 nm, 10 absorbance units (AU) $mM^{-1}$ $cm^{-1}$), 4-thio-Ura (**15**) (270 nm, 22 AU $mM^{-1}$ $cm^{-1}$), 2-thio-Ura (**16**) (273 nm, 12 AU $mM^{-1}$ $cm^{-1}$), thymidine (**20**) (260 nm, 8.9 AU $mM^{-1}$ $cm^{-1}$), guanine (**28**) (246 nm, 10.7 $mM^{-1}$ $cm^{-1}$), cytosine (**23**) (267 nm, 6.1 $mM^{-1}$ $cm^{-1}$), adenine (**27**) (260 nm, 13 $mM^{-1}$ $cm^{-1}$).

**High-performance liquid chromatography (HPLC).** Nucleosides were quantified by reversed-phase ion-pairing HPLC. Typically, 5 μL of sample containing an overall nucleoside concentration of about 1 mM were loaded on a Kinetex C18 column (Phenomenex, Aschaffenburg, Germany; 5 μm, 100 Å, 50 × 4.6 mm). Analytes were separated in 5-min long isocratic runs using 20 mM phosphate buffer, pH 5.9, containing 40 mM tetra-n-butylammonium bromide (TBAB) and 12.5% acetonitrile. The flow rate was 2 mL/min and the temperature set to 35 °C. Eluents were detected at 260 nm. A 5-min HPLC trace is shown in Supplementary Fig. 35. Typical retention times were as follows: Ura (**12**) (0.3 min), ΨMP (0.4 min), ΨDP (0.9 min), ADP (1.1 min), ΨTP (2.7 min), and ATP (4 min).

**Thin-layer chromatography (TLC).** About 1–2 μL sample were spotted onto a TLC plate (Merk, Darmstadt, Germany). The mobile phase was composed of 2-BuOH:AcOH:$H_2O$ (2:1:1). The development solution consisted of 0.5 g thymol, 95 mL EtOH and 5 mL $H_2SO_4$. Reducing sugar spots were developed by heating the plate to approximately 70 °C for 2 min using a heat-gun (Supplementary Fig. 33).

**Synthesis and isolation of pentose 5-phosphates.** Prior to synthesis, the substrate scope of RibK was evaluated using mixture of 100 mM sugar (D-Rib, D-dRib, D-Xyl, D-Ara, 2F-D-Rib, D-Glc, 2F-D-Rib, 3A-D-Rib, L-Xyl, L-Ara, D-Lyx, D-Man, D-GlcN, D-GlcNAc, D-GlcA, D-Gal, D-GalN), 100 mM PEP, 2 mM ATP and 10 mM HEPES buffer (pH 7.0) containing 1 mM $MgCl_2$ and enzymes (60 μM RbsK, 0.4 μM PK) in a total volume of 100 μL. The reactions were incubated at 37 °C without agitation for 20 h and 5 μL sample were analyzed by TLC. Of the tested sugars, D-Rib, D-dRib, D-Xyl, D-Ara, 2F-D-Rib, D-Glc were accepted.

Synthesis was performed at 5 mL scale, except for 2-deoxy-2-fluoro-D-Rib 5-phosphate (2F-D-Rib5P (**5**) that was prepared in a 1 mL reaction. Reaction mixtures contained 100 mM pentose (D-Rib, D-dRib, D-Xyl, D-Ara, 2F-D-Rib), 100 mM PEP, 2 mM ATP in 10 mM HEPES buffer (pH 7.0) with 1 mM $MgCl_2$ and 0.4 μM PK. For the phosphorylation of D-Rib, 15 μM RbsK, for D-dRib 30 μM RbsK, and for other pentoses 60 μM RbsK were used. The reactions were incubated at 37 °C without agitation for 20 h and 5 μL sample were analyzed by TLC. The reaction was stopped by heat treatment at 99 °C for 10 min and precipitated protein was removed by centrifugation for 10 min at 4 °C and 16,000 × *g*. The reactions were diluted with twice volume of water and applied (20 mL; 2 mL/min) to a FliQ FPLC column (10 mL; 104 × 11.0 mm; Generon, Maidenhead, U.K.) packed with SuperQ-650M (Tosh, Tokyo, Japan) anion exchange resin mounted onto an ÄKTA prime plus FPLC system (GE Healthcare). The column was equilibrated in 10 mM sodium acetate (NaAc), pH 4.5. Products were eluted using a linear gradient of 1 M NaAc (pH 4.5), from 0% to 50% in 120 mL with a flow of 4 mL/min. Eluting fractions were evaluated for the presence of sugar phosphates using TLC. Following the product elution, the column was regenerated with 100% of 1 M NaAc buffer for 10 min. The fractions containing pentose 5-phosphates were pooled and concentrated under reduced pressure (20 mbar, 40 °C) to a total volume of 1 mL using a rotary evaporator (Laborta 500-efficient; Heidolph Instruments, Schwabach, Germany). The concentration of pentose 5-phosphates were evaluated as follows: sugar phosphates were completely hydrolyzed by applying 10 μM of sugar phosphate phosphatase (AGP)[88] for 20 h. The released inorganic phosphate was determined in a colorimetric assay based on the formation of a blue phospho-molybdate complex at 850 nm[89].

**Screening sugar phosphate and nucleobase substrates.** All reactions were performed at 37 °C in 50 mM HEPES buffer (pH 7.0) containing 2 mM $MnCl_2$ in a total volume of 100 μL. Nucleobase screening was performed by incubating 5 mM D-Rib5P (**1**) and 1 mM of nucleobase (Ura (**12**), 3-methyl-Ura (**13**), 6-amino-Ura (**14**), 2-thio-Ura (**15**), 4-thio-Ura (**16**), 1-methyl-Ura (**17**), 5-fluoro-Ura (**18**), 6-trifluoro-methyl-Ura (**19**), 6-aza-Ura (**21**), 6-Cl-Ura (**22**), 2,4-dithio-Ura (**24**), 5-ethynyl-Ura (**25**), maleimide (**26**), cytosine(**23**), thymine (**20**), adenine (**27**), guanine (**28**), 4-pyrimidone (**29**) and pyrimidine (**30**)) with 30 μM YeiN. Sugar phosphates were screened by incubating 1 mM Ura and 5 mM of sugar phosphate (D-Rib5P (**1**), D-dRib5P (**2**), D-Xyl5P (**3**), D-Ara5P (**4**), D-Fru6P (**9**), D-Man6P (**8**), D-Glc6P (**7**), D-Rul (**10**), D-Ery4P (**11**), 2F-D-Rib5P (**5**), L-Rib5P (**6**) with 30 μM YeiN. Samples (20 μL) were withdrawn after 0, 3, 24 h and quenched by methanol addition (100 μL). Precipitated protein was removed by centrifugation at 20,000 × *g* for 5 min and supernatant was analyzed by HPLC.

Five millimolar of pentose phosphates (previously isolated as described above) accepted by YeiN in the initial screen (D-Rib5P (**1**), D-dRib5P (**2**), D-Xyl5P (**3**), D-Ara5P (**4**), D-2F-Rib5P(**5**)) were incubated with 2 mM of each nucleobase (Ura (**12**), 3-methyl-Ura (**13**), 6-amino-Ura (**14**), 2-thio-Ura (**15**), 4-thio-Ura (**16**), 1-methyl-Ura (**17**), 5-fluoro-Ura (**18**), 6-trifluoro-methyl-Ura (**19**), 6-aza-Ura (**21**), 6-Cl-Ura (**22**), 2,4-dithio-Ura (**24**), 5-ethynyl-Ura (**25**), maleimide (**26**), cytosine (**23**), thymine (**20**), adenine (**27**), guanin (**28**), 4-pyrimidone (**29**) and pyrimidine

(**30**)) and 30 μM of YeiN for 3 h. The reactions were quenched by methanol addition (500 μL) and precipitated protein was removed by centrifugation at 20,000 × $g$ for 5 min prior to HPLC analysis.

**YeiN kinetics**. The rate constant $k_{cat}$ and the Michaelis constant $K_M$ were determined by measuring the initial rate of ΨMP synthesis catalyzed by YeiN (1 μM–120 μM) in 50 mM HEPES (pH 7.0) supplemented with 2 mM MnCl₂. Reactions were incubated at 37 °C on a Thermomixer comfort (Eppendorf, Hamburg, Germany) with agitation of 300 rpm. Initial rates for ΨMP synthesis with nucleobases (Ura (**12**), 6-amino-Ura (**13**), 4-thio-Ura (**14**), 2-thio-Ura (**15**), 3-methy-Ura(**16**)) were determined at a constant D-Rib5P (**1**) concentration of 5 mM and at nucleobase concentrations between 0.05 mM and 1.00 mM. Initial rates for ΨMP synthesis with pentose 5-phosphates (D-Rib5P (**1**), D-dRib5P (**2**), D-Ara5P (**4**), D-Xyl5P(**3**)) were determined at a constant Ura (**10**) concentration of 5 mM and 1.00−0.05 mM of the corresponding pentose 5-phosphate. Reactions were started by YeiN (1.00 μM−100 μM) added in a concentration adjusted to the enzyme's specific activity with the substrate used. The reaction mixture was incubated for 20 min and quenched by methanol addition (1:1, v:v). Precipitated protein was removed by centrifugation at 20,000 × $g$ for 5 min and samples were analyzed by HPLC. It was ensured that conversion was below 20%. Kinetic parameters were obtained from non-linear least squares fits (SigmaPlot; Systat, Erkrath, Germany) of initial ΨMP-synthesis rates to Michaelis–Menten Eq. (1). $V$ is the initial rate, $V_{max}$ is the maximum initial rate, $K_M$ is the Michaelis constant and [S] is the initial substrate concentration. The $k_{cat}$ was determined from the relationship $k_{cat} = V_{max}/$ [E], where [E] is the molar enzyme concentration. [E] was determined from the protein concentration and enzyme molecular weight.

$$V = \frac{V_{max}[S]}{(K_M + [S])} \quad (1)$$

**Synthesis of ΨMP derivatives**. Synthesis of ΨMP derivatives was performed in two steps in one pot. Reactions were started by addition of enzyme. All reactions were incubated at 37 °C without agitation.

Step one: pentose phosphorylation. The reaction mixture contained 100 mM pentose (D-Rib, D-dRib, D-Xyl, D-Ara, 2F-D-Rib), 100 mM PEP, 2 mM ATP and 10 mM HEPES buffer (pH 7.0), 1 mM MgCl₂ and 0.4 μM pyruvate kinase. For the phosphorylation of D-Rib 15 μM RbsK, for D-dRib 30 μM RbsK and for D-Ara and D-Xyl 60 μM RbsK was used. Reactions were performed in a total volume of 5 mL. Samples (5 μL) were taken at reaction start and then after 2 h and 20 h. They were analyzed by TLC.

Step two: C-glycosylation using YeiN. After phosphorylation, the reaction mixture was diluted to a final concentration of 25 mM pentose 5-phosphate (D-Rib5P (**1**), D-dRib5P (**2**), D-Xyl5P (**3**), D-Ara5P(**4**)) using 10 mM HEPES (pH 7.0). Ten mM of nucleobase (Ura (**12**), 6-amino-Ura (**13**), 2-thio-Ura (**14**), 4-thio-Ura (**15**), 3-methyl-Ura (**16**)) were added to the phosphorylation reaction. ΨMP-synthesis was started with 15 μM YeiN. To compensate for the lower activity of YeiN towards different pentose 5-phosphates and nucleobases, the YeiN concentration was adjusted as follows: for the synthesis of dΨMP 30 μM YeiN was used; for synthesis of Ara-ΨMP, Xyl-ΨMP, 6-amino-ΨMP, 2-thio-ΨMP and 4-thio-ΨMP, 90 μM YeiN was used. Samples (5 μL) were taken at start and after 5, 15, 30, 60, 120, 180, 240, 300, 360 min, and 10 h for 2-thio-ΨMP, Ara-ΨMP, Xyl-ΨMP. Each sample was quenched by 100 μl methanol. Precipitated protein was removed by centrifugation for 20,000 × $g$ for 5 min and the sample was analyzed by HPLC. After completion of the reaction or latest after 10 h of incubation, the reactions were stopped by heat inactivation for 5 min at 99 °C. Precipitated protein was removed by centrifugation for 10 min at 4 °C and 16,000 × $g$.

**ΨMP dephosphorylation**. ΨMP (5 mM) was dissolved in 15 mL 10 mM HEPES buffer, pH 8.0, supplemented with 1 mM MgCl₂ and incubated with 2 U/mL of CIP at 37 °C for 600 min. Ten microliters samples were withdrawn at start, and after 30, 90, 600 min, and quenched in 100 μL methanol. Precipitated enzyme was removed by centrifugation (16,000 × $g$, 4 °C, 5 min) and supernatant was analyzed by HPLC.

**CMPK, NDK, PK, phosphorylation cascade**. The reaction set-up consisted of 0.5 mM ATP, 5 mM purified ΨMP, 12 μM CMPK, 1 mM MgCl₂, 30 μM PK, 12 mM PEP, 50 mM HEPES, pH 7.0, in a total volume of 250 μL. Ten μL samples taken at certain times were quenched in 100 μL methanol. The precipitated enzyme was removed by centrifugation (20,000 × $g$, 4 °C, 5 min) and supernatant was analyzed by HPLC. To test the effect of NDK removal for ΨTP-synthesis the same reaction set up without addition of 40 μM NDK was used (Supplementary Fig. 31).

**One-pot multistep enzymatic synthesis of Ψ-triphosphates**. Pentose phosphorylation was performed as described above under "Step one: pentose phosphorylation". For ΨMP synthesis, 15 mM nucleobase (Ura (**12**), 6-amino-Ura (**13**), 2-thio-Ura (**14**), 4-thio-Ura (**15**), 3-methyl-Ura (**16**)), 1 mM MnCl₂, and 10 mM HEPES buffer were added to the reaction mixture to a final concentration of 25 mM pentose 5-phosphate. The reaction was started with YeiN (1 μM for the ΨMP synthesis, 3 μM for the dΨMP synthesis and 90 μM for the synthesis using Ura derivatives). ΨMP phosphorylation was started with 15 mM PEP, 15 μM

CMPK, and 1 mM ATP, resulting in a volume increase of 33%. Ten microliters of samples were quenched by addition of 100 μL methanol and precipitated protein was removed by centrifugation for 20,000 × $g$ for 5 min prior to HPLC analysis. Enzymes were heat inactivated (5 min, 99 °C) after 21 h reaction and precipitated protein was removed by centrifugation for 10 min at 4 °C and 16,000 × $g$.

**Dephosphorylation of pentose 5-phosphates**. BT4131 (10 μM) and 1 mM MgCl₂ were added to the reaction mixtures and incubated for 60 min of at 37 °C. The reaction was stopped by heat inactivation (5 min, 99 °C) and precipitated protein was removed by centrifugation for 10 min at 4 °C and 16,000 × $g$.

**Anion exchange chromatography**. Enzyme-free reaction mixture was applied (20 mL; 2 mL/min) to a FliQ FPLC column (10 mL; 104 × 11.0 mm; Generon, Maidenhead, U.K.) packed with SuperQ-650M (Tosh, Tokyo, Japan) anion exchange resin mounted onto an ÄKTA prime plus FPLC system (GE Healthcare.). The column was equilibrated in 10 mM NaAc, pH 4.5. Products were eluted using a linear gradient of 1 M NaAc, pH 4.5, from 0% to 50% in 120 mL with a flow of 4 mL/min and monitored at 260 nm. Following product elution, buffer B was set to 100% for 10 min to elute unspecifically bound substances. The fractions containing the desired C-nucleoside were pooled and concentrated under reduced pressure (20 mbar, 40 °C) to a total volume of 10 mL using a rotary evaporator (Laborta 500-efficient; Heidolph Instruments, Schwabach, Germany).

**Size exclusion chromatography**. To remove excess NaAc, concentrated products were split in 5 × 2 mL portions and loaded onto a size exclusion column (16 × 1000 mm; XR 16/100 column; GE Healthcare) packed with Sephadex G10 (exclusion limit <700 Da) equilibrated in double distilled H₂O and mounted onto an ÄKTA prime plus FPLC system (GE Healthcare). Compound elusion was performed using doubly distilled water and monitored at 260 nm. Fractions containing salt-free product were collected, pooled and concentrated under reduced pressure (20 mbar, 40 °C) to a total volume of 10 mL using a rotary evaporator (Laborta 500-efficient; Heidolph Instruments) and lyophilized.

**Docking**. Molecular docking was used to structurally evaluate the substrate scope of YeinN. AutoDock13[90] as implemented in Yasara v. 18.2.7 was used for enzyme-ligand docking. The AmberFB-15[91] force field and the default parameters provided by the standard docking macro were used. The structure of YeiN (pdb code: 4GIL)[62] was used as a receptor in molecular-docking experiments. For dockings of sugar phosphates (D-Rib5P (**1**), D-Rul5P (**10**), D-Fru6P (**9**), D-Glu6P (**7**), D-Ery4P(**11**)), the ring-opened intermediate was deleted and a simulation cell of 15 Å × 10 Å × 10 Å was placed at the former location of D-Rib5P (**1**). Sugar phosphates were docked in the ring-open conformation. The protonation state of key catalytic residues (Glu31 protonated; Lys166 deprotonated) was set prior to the docking experiment. The protonation states of all other protein residues were set automatically for a pH of 7.00. For dockings of nucleobases (Ura (**12**), 6-trifluoro-methyl-Ura (**19**), 6-Cl-Ura (**22**), 6-aza-Ura (**21**), 1-methyl-Ura (**17**), 5-fluoro-Ura (**18**), 5-methyl-Ura(**20**)), the uracil moiety was deleted and the charge of lysine was set to fit the covalent iminium intermediate. The protonation states of all other protein residues were set automatically for a pH of 7.00. A simulation cell of 15 Å × 10 Å × 10 Å was placed at the former location of Ura (**11**). Following the docking runs, the obtained poses were optimized by energy using the standard macro included in Yasara. All ligands were generated using the Grade Web Server (http://grade.globalphasing.org/cgi-bin/grade/server.cgi)[92]. Docking poses were evaluated by their associated free energy and mechanistic plausibility.

**Synthesis of pseudouridinated RNA**. In vitro translation (IVT) using T7 RNA polymerase (Thermo Fisher Scientific) was performed following the manufacturer's protocol. Shortly, linear DNA template containing the T7 promoter and the yein gene was amplified from the YeiN expression vector (peT15_yein) by PCR using Phusion polymerase and a set of primers complementary to the T7 promoter (TAATACGACTCACTATAGGG) or terminator (GCTAGTTATTGCTCAGCGG). The PCR product was purified via column purification (GenJET® PCR clean up; Thermo Fisher Scientific) and eluted in DEPC-water. IVT was performed using 30 U of T7 RNA polymerase, 600 ng template DNA, Transcription Buffer (40 mM Tris-HCl pH 7.9, at 25 °C), 6 mM MgCl₂, 10 mM DTT, 10 mM NaCl and 2 mM spermidine) and 2 mM of each NTP (ATP, GTP, UTP, CTP) in a total volume of 50 μL. Synthesis of modified RNA was performed in two ways; either 2 mM of ΨTP (or: dΨTP, 6-amino-ΨTP) was additionally added to the conventional NTPs or, UTP was fully replaced by 2 mM of either ΨTP, dΨTP or 6-amino-ΨTP. The reaction was incubated at 37 °C in the thermocycler (Doppio-Dual; VWR, Radnor, PA, USA) and the resulting PCR products were evaluated by agarose gel electrophoresis.

**Synthesis of pseudouridinated DNA**. To evaluate the potential of PCR catalyzed generation of pseudouridinated DNA, the yein gene was amplified from 50 ng expression vector (pet15_yein) using 1.25 U of Dream-Taq DNA polymerase, 0.2 mM of dNTPs (dATP, dCTP, dGTP, dTTP), 0.5 μM T7_fwd, 0.5 μM T7_bwd primers and DreamTaq Buffer as recommended by the supplier (Thermo Fisher

Scientific). To incorporate Ψ into the DNA either 0.2 mM of ΨTP were additionally added to dNTP mix or fully replaced dTTP by dΨTP. The temperature profile consisted of a preheating step at 95 °C for 1 min, followed by 30 reaction cycles of denaturation at 95 °C for 30 s, annealing at 55 °C for 30 s, and elongation at 72 °C for 2 min. The final extension step was carried out at 72 °C for 5 min. PCRs were performed using the Doppio-Dual thermocycler (VWR). The resulting PCR products were evaluated by agarose gel electrophoresis.

**In situ [1]H-NMR of ΨMP synthesis or hydrolysis**. YeiN-catalyzed synthesis of ΨMP from D-Rib5P (**1**) and Ura (**12**), as well as hydrolysis of ΨMP, were analyzed by [1]H-NMR. All reagents including the enzyme were dissolved in $D_2O$ and the reaction was set up in a 5 mm high-precision NMR sample tube (Promochem, Wesel, Germany). YeiN was buffered by washing the enzyme with 10-fold excess of 10 mM potassium phosphate buffer (pD 7.5) supplemented with 0.1 mM $MnCl_2$, using centrifugation (16,000 × g, 15 min, 4 °C) with Vivaspin 500 centrifugal concentrators (10 kDa cut-off). The synthesis reaction contained 14 mM Ura (**12**) and 8 mM D-Rib5P (**1**), while 10 mM ΨMP were used for the hydrolysis reaction. Each reaction was initiated by adding 3 μM enzyme. The reactions were incubated at 30 °C without agitation in the electromagnetic field.

Measurements were performed with a Varian INOVA 500-MHz NMR spectrometer (Agilent Technologies; Santa Clara, CA, USA). The VNMRJ 2.2D software was used for all measurements. [1]H-NMR spectra (499.98 MHz) were recorded on a 5 mm indirect detection PFG-probe with pre-saturation of the water signal by a shaped pulse. The following standard pre-saturation sequence was used: 2 s relaxation delay; 90° proton pulse; 2.048 s acquisition time; 8 kHz spectral width; number of points 32 k.

**[1]H-NMR analysis of YeiN-catalyzed deuterium exchange**. The potential of YeiN-catalyzed deuterium wash-in into Ura (**12**) without transient formation of ΨMP was evaluated by [1]H-NMR. All reagents including the enzyme were dissolved in $D_2O$ and prepared as described above. The 600 μL reactions contained 5 mM Ura (**12**) and 5 mM of sugar phosphate (D-Rib5P (**1**), D-Ery4P (**11**), D-Rul5P (**10**), D-Glc6P (**7**), D-Fru6P(**9**)) in 10 mM potassium phosphate buffer (pD 7.5) supplemented with 0.1 mM $MnCl_2$. Conversions were started by addition of 3 μM enzyme and incubated at 30 °C in a Thermomixer comfort (Eppendorf, Hamburg, Germany) with agitation of 300 rpm for 3 h. The reactions were stopped by heat treatment (5 min, 99 °C) and precipitated protein was removed by centrifugation for 10 min at 4 °C and 16,000 × g. The supernatant was transferred into a 5 mm NMR sample tube and analyzed by [1]H-NMR.

**NMR analysis of purified reaction products**. Lyophilized products were dissolved to 3–10 mM in 50 mM potassium phosphate buffer (pD 7.9) in a total volume of 600 μL and transferred to a 5 mm high-precision NMR sample tube. They were analyzed using a Varian INOVA 500-MHz NMR spectrometer (Agilent Technologies). The VNMRJ 2.2D software was used for measurements. [1]H-NMR spectra (499.98 MHz) were measured on a 5 mm indirect detection PFG probe. Standard pre-saturation sequence was used: relaxation delay 2 s; 908 proton pulse; 2.048 s acquisition time; spectral width 8 kHz; number of points 32,000 13C NMR spectra (125.71 MHz). Standard pre-saturation sequence was used: relaxation delay 2 s; 908 proton pulse; 2.048 s acquisition time; spectral width 8 kHz; number of points 32 k. MNova 12.0.3 (Mestrelab Research S.L. Bajo, Santiago de Compostela, Spain) was used for data processing and analysis (Supplementary Figs. 10–29). Recorded spectra were in good agreement with published data[75].

**Reporting summary**. Further information on research design is available in the Nature Research Reporting Summary linked to this article.

## Data availability
All relevant data are reported in the manuscript and in the associated Supplementary Information. All data are available from the corresponding author upon reasonable request. There is no restriction on data availability. Source data are provided with this paper.

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

## Acknowledgements

We thank Alma Memic and Stefan Drießler (Institute of Biotechnology and Biochemical Engineering, Graz University of Technology) for experiments; Prof. Hansjörg Weber (Institute of Organic Chemistry, Graz University of Technology) for NMR

measurements. Financial support from the Austrian Science Funds (FWF, DK Molecular Enzymology W901; to B.N. and M.P.) is gratefully acknowledged.

## Author contributions

M.P. design of study; all experiments and data analysis; writing of paper. B.N. design of study; funding acquisition; discussion; writing of paper.

## Competing interests

The authors declare no competing interests.
