## [Peer Review File · Nature Communications]

REVIEWER COMMENTS

Reviewer #1 (Remarks to the Author):

Pfeiffer and Nidetzky have presented a well-written, clear and elegant study that demonstrate the use of a C-glycosidase to generate a range of C-nucleotides.

This work is scientifically and technically relevant, and is innovative in the corresponding scientific fields, as it describes a simple biocatalysis pathway to synthesis the most useful C-nucleosides.

Therefore, I would recommend publication in Nature Communications after taking into account the following revisions and comments.

Minor revisions and comments :

- Figure 1: Why Glu3 is not indicated as the Acid/base catalytic residue in YeiN mechanism, as it is described as such in the conclusion ?

- Figure 2: the numbering of uridine analogues is wrong (maleimide), and some structures are missing (cytosine). Please correct.

- Line 144: is the proposed mechanism is close to those reported for aldolases, what is the sequence or structural homology of YeiN with aldolases ?

- Line 173-175: AlnA does not produce a unique anomer after C-glycosylation. Is it because of the cyclization that occurs in slution (unlike YeiN) ? This should be clarified in the manuscript, as it is confusing for readers.

- Figure 4: Please use the same scale for NMR spectra, which will enable easier comparison between both experiments. I do not understand figure 4f of Ψ MP hydrolysis, as Rib5P appearance is indeed correlated to 5-D-Ura formation. Yet, Rib5Pa and Rib5Pb are constant and present from the beginning of the reaction. Is if proportions of each anomer of the Rib5P formed ? This should be clarified.

- Figure 6: The 50% decrease of band intensity when using Ψ TP instead of UTP in RNA polymerase reaction is not obvious. Or the manuscript text is unclear, as it points out the reaction when Ψ TP is added to standard NTP mixture (with UTP). In this cas, the text should be rewritten for clarity.

- lines 272/273. Fig6 clearly shows that d Ψ TP cannot be incorporated by DreamTaq when it is replaing dTTP. Yet, the authors claim that when adding th C-nucleotide to the dNTP mixture, d Ψ TP is inhibiting the reaction (which can be understood), but also that it might be incorporated in low % (and thus could be detected). This claim should be clarified, as it is not easily understandable as such.

Review made by Dr. P. Lafite, Orleans, France

Reviewer #2 (Remarks to the Author):

The paper of Pfeiffer et al describes the utility of pseudouridine monophosphate C-glycosidase for selective 5-beta-C-glycosylation of uracil and derivatives from pentose 5-phosphate substrates. They also set up a cascade reaction for the phosphorylation of both the starting material and the resulting C-nucleoside. The paper is innovative although the enzyme was discovered several years ago, but not its synthetic potentiality in the synthesis of C-nucleosides. The paper contains interesting information that can be useful in the synthetic community both from the catalyst and

the products synthesized. Experimentally the paper is well done and easy to follow. In summary, it can be acceptable in Nature Communications

However some specific points must be taken into consideration:

a) The authors stated that the enzymatic process is a C-C aldol condensation. However examined the reaction in more detail, it looks like a Mannich-like addition (addition of a nucleophile to an iminium ion) with elimination of the amine of the Lys followed by an oxa-Michael addition. Could you please check if this opinion agrees with Figure 1 and that from the supplementary information?

b) In the substrate scope, the authors only combine Ura with a series of sugar phosphates and nucleobase analogues in combination with Rib5P. It would be interesting to combine with other sugars and other nucleobases as well, because in an enzymatic bimolecular process the donor and acceptor affects the tolerance of each other.

c) The mechanistic analysis of the Yein reaction is, in the opinion of this referee, the weakest part of the paper. Actually, this section consists of a description of molecular docking results that give an unprecise idea of the mechanism and does not explain well the role of the residues in each catalytic step. The lack of QM/MM study is the cause of this deficiency.

In this section the authors stated that: "Contrary 145 to Rib5P, the docking poses for Ery4P, Ru15P and Fru6P have Thr139 and Glu31 not properly aligned to the carbonyl oxygen (Figure 3b-e)." Comparisons with Rib5P (DERA) is misleading. This is because the Lysine in DERA directs the formation of active nucleophile, i.e. enamine and, in this case, directs an electrophilic carbon activation (a type of Mannich acceptor).

In another paragraph: "The Glc6P pose shows an enlarged distance 147 between the epsilon-amino of Lys166 and the C1 carbon. Lack of activity of these sugar phosphates is thus plausibly explained." What was the situation with the other less sterically demanding sugar number 8?

And: "Due to electronic effect, the fluorine and the trifluoromethyl substituents at C5 and C6, respectively,..." if the binding site is narrow it must be also steric effects. Please discuss that. This is in fact similar to the 1-methyl-Ura.

d) Table 1: For those unfamiliar with the nucleoside/nucleotide nomenclature it would be very helpful to show the structures too.

Revision

Author responses are in light blue. Changes in the manuscript are marked in yellow background. The Supplementary Information was revised as necessary. Changes are described in the individual responses. A “clean” copy of the Supplementary Information is uploaded. The editorial requirements have been considered. A data source file is uploaded.

REVIEWER COMMENTS

Reviewer #1 (Remarks to the Author):

Pfeiffer and Nidetzky have presented a well-written, clear and elegant study that demonstrate the use of a C-glycosidase to generate a range of C-nucleotides.

This work is scientifically and technically relevant, and is innovative in the corresponding scientific fields, as it describes a simple biocatalysis pathway to synthesis the most useful C-nucleosides.

Therefore, I would recommend publication in Nature Communications after taking into account the following revisions and comments.

Response: We thank the reviewer for positive overall assessment of the manuscript.

Minor revisions and comments :

- Figure 1: Why Glu3 is not indicated as the Acid/base catalytic residue in YeiN mechanism, as it is described as such in the conclusion ?

Response: We have added Glu31 to Figure 1 and describe its participation in the proposed catalytic mechanism. There are steps in the enzymatic mechanism (e.g., sugar ring opening/closure) where it is not clear whether an enzyme residue is involved catalytically; and if so, which residue of the YeiN it is. In that case, we retained the usage of B: for unknown enzyme base and A-H for unknown enzyme acid.

- Figure 2: the numbering of uridine analogues is wrong (maleimide), and some structures are missing (cytosine). Please correct.

Response: The numbering was corrected and a complete list of compounds is now shown. During revision, a number of additional sugar phosphates and nucleobases were tested. These compounds are now also added.

- Line 144: is the proposed mechanism is close to those reported for aldolases, what is the sequence or structural homology of YeiN with aldolases ?

Response: Reviewer #2 pointed to the fact that our discussion of the structural and mechanistic relatedness of YeiN with aldolases was not clear enough. We agree with Reviewer #2 and have therefore added discussion that now draws clear line between what is similar/distinct between YeiN and aldolases. Being clear about the difference between the two types of enzyme is mechanistically important. The sequence identity between YeiN and aldolases is hardly significant. However, the arrangement of active-site residues involved in

the formation of the iminium ion intermediate by YeiN is highly similar to the one in aldolases. The expanded mechanistic discussion makes these points clear.

- Line 173-175: AlnA does not produce a unique anomer after C-glycosylation. Is it because of the cyclization that occurs in solution (unlike YeiN) ? This should be clarified in the manuscript, as it is confusing for readers.

Response: We clarify the discussion. It is believed that AlnA reaction involves cyclization in solution. A key paper (Oja, L. *Proc. Natl. Acad. Sci.* **2013**, *110*, 1291–1296) emphasizes this important point. We have changed Figure S3 to show cyclization in solution and have modified the main text.

- Figure 4: Please use the same scale for NMR spectra, which will enable easier comparison between both experiments. I do not understand figure 4f of Ψ MP hydrolysis, as Rib5P appearance is indeed correlated to 5-D-Ura formation. Yet, Rib5Pa and Rib5Pb are constant and present from the beginning of the reaction. Is if proportions of each anomer of the Rib5P formed ? This should be clarified.

Response: We have modified Figure 4 for improved readability. The scales of the NMR spectra in panel b) and e) have been corrected and aligned. An inset was used to emphasize the NMR signals from alpha and beta anomeric protons of Rib5P. In the time course, we used a second axis to show the formation of the alpha and beta anomers of Rib5P. The mutarotation of Rib5P is too fast to be monitored with the NMR methods used. It is therefore that the alpha/beta forms always appear at equilibrium.

- Figure 6: The 50% decrease of band intensity when using Ψ TP instead of UTP in RNA polymerase reaction is not obvious. Or the manuscript text is unclear, as it points out the reaction when Ψ TP is added to standard NTP mixture (with UTP). In this cas, the text should be rewritten for clarity.

Response: We agree and have rewritten the relevant part of the manuscript. The text now reads as follows. Line 300-304 "*Densitometric analysis of gel bands suggested that RNA polymerase transcription efficiency (expressed as gel band intensity) was similar when Ψ TP was used to substitute UTP. Unexpectedly, the band intensity decreased to about half when Ψ TP and UTP were added at the same time. Overloading of the RNA polymerase with NTP substrate may have caused inhibition, perhaps due to limitation in the concentration of Mg^{2+} used.*"

- lines 272/273. Fig6 clearly shows that d Ψ TP cannot be incorporated by DreamTaq when it is replaing dTTP. Yet, the authors claim that when adding th C-nucleotide to the dNTP mixture, d Ψ TP is inhibiting the reaction (which can be understood), but also that it might be incorporated in low % (and thus could be detected). This claim should be clarified, as it is not easily understandable as such.

Response: We have rewritten the text for clarity. The statement that the d Ψ TP may have been incorporated was speculative. It was deleted.

Review made by Dr. P. Lafite, Orleans, France

Reviewer #2 (Remarks to the Author):

The paper of Pfeiffer et al describes the utility of pseudouridine monophosphate C-glycosidase for selective 5-beta-C-glycosylation of uracil and derivatives from pentose 5-phosphate substrates. They also set up a cascade reaction for the phosphorylation of both the starting material and the resulting C-nucleoside. The paper is innovative although the enzyme was discovered several years ago, but not its synthetic potentiality in the synthesis of C-nucleosides. The paper contains interesting information that can be useful in the synthetic community both from the catalyst and the products synthesized. Experimentally the paper is well done and easy to follow. In summary, it can be acceptable in Nature Communications

However some specific points must be taken into consideration:

Response: We thank the reviewer for positive overall assessment of the manuscript.

a) The authors stated that the enzymatic process is a C-C aldol condensation. However examined the reaction in more detail, it looks like a Mannich-like addition (addition of a nucleophile to an iminium ion) with elimination of the amine of the Lys followed by an oxa-Michael addition. Could you please check if this opinion agrees with Figure 1 and that from the supplementary information?

Response: We agree with the reviewer and are grateful for an excellent mechanistic suggestion which is in full accordance with our own thinking of the enzymatic reaction. We were initially unsure as to the details of the mechanistic description that could be used. We feel encouraged by the comment of the reviewer and have expanded the discussion in the main text. The key feature, that the C-C coupling can be considered to proceed according to a Mannich-like reaction is mentioned in the Abstract, in the legend to Figure 1 and in particular later under Results and Discussion. Considering that we do not have very detailed evidence on each step of the enzymatic mechanism, our mechanistic discussion is still somewhat tempered to avoid speculation.

b) In the substrate scope, the authors only combine Ura with a series of sugar phosphates and nucleobase analogues in combination with Rib5P. It would be interesting to combine with other sugars and other nucleobases as well, because in an enzymatic bimolecular process the donor and acceptor affects the tolerance of each other.

Response: We agree with the reviewer. During revision, we have expanded the list of sugar phosphates and nucleobase analogues tested. We also have performed a combinatorial analysis of reaction of sugar phosphates and nucleobase analogues. The substrate specificity of the enzyme is thus characterized better than it was in the original version of the manuscript. However, the specificity of the enzyme is confirmed fully.

c) The mechanistic analysis of the Yein reaction is, in the opinion of this referee, the weakest part of the paper. Actually, this section consists of a description of molecular docking results that give an unprecise idea of the mechanism and does not explain well the role of the residues in each catalytic step. The lack of QM/MM study is the cause of this deficiency.

Response: We have adopted the mechanistic suggestion of the reviewer and feel that we now provide a much better and more coherent description of the mechanism. We believe that the criticism of the reviewer at least in part arose due to the way we have drawn analogy to the aldolase reaction. We realized that our description may have been

misleading. Despite some analogies present between YeiN and aldolases concerning the formation of the covalent iminium intermediate, it is important to make clear distinction between the two enzymes mechanistically at subsequent stages of the reaction. This is what we have done in the revised version of the manuscript. In addition, we have presented the results of the docking study in a clearer manner. We also avoid the drawing of mechanistic conclusions that a docking analysis cannot support.

We agree with the reviewer that it would be interesting to perform a QM/MM analysis. However, a full-fledged computational study on the enzymatic reaction with its multiple catalytic steps would be a different study and would likely have to be a scientific account on its own. In terms of methodology we wouldn't be able to perform that research. Despite these possible limitations, we are of the opinion that a QM/MM analysis is not a support that the current study requires to stand by itself. Our response to the critique of the reviewer is to improve the mechanistic presentation. We also renamed the section to "Characterization of the enzymatic reaction" to avoid the impression that we have fully clarified the enzymatic mechanism.

In this section the authors stated that: "Contrary to Rib5P, the docking poses for Ery4P, Rul5P and Fru6P have Thr139 and Glu31 not properly aligned to the carbonyl oxygen (Figure 3b-e)." Comparisons with Rib5P (DERA) is misleading. This is because the Lysine in DERA directs the formation of active nucleophile, i.e. enamine and, in this case, directs an electrophilic carbon activation (a type of Mannich acceptor).

Response: We fully agree with the reviewer. The relevant section was rewritten to make clear the difference between YeiN and aldolases such as DERA. We have also clarified where in the mechanism (formation of the iminium ion intermediate), and in which way (stabilizing residues), we find analogy between YeiN and aldolases. We furthermore show that in the corresponding docking poses, sugar phosphates such as Ery4P, Rul5P and Fru6P are not well positioned to recruit stabilization from the enzyme residues to facilitate formation of the covalent intermediate. We agree with the reviewer on the limitations of docking analysis. We would like to emphasize therefore that the docking results are only used to provide an interpretation of what the experiments have already shown. We believe that used thus, the docking is a useful complement.

In another paragraph: "The Glc6P pose shows an enlarged distance 147 between the epsilon-amino of Lys166 and the C1 carbon. Lack of activity of these sugar phosphates is thus plausibly explained." What was the situation with the other less sterically demanding sugar number 8?

Response: We show the docking poses for relevant sugar phosphates in the Supplementary Information (Figure S8). Positioning appears to be poor, or not good enough for catalysis, in Ery4P, Rul5P, Fru6P and Glc6P. We think that the docking poses are clear and deleted the sentence that "lack of activity of these sugar phosphates is thus plausibly explained".

And: "Due to electronic effect, the fluorine and the trifluoromethyl substituents at C5 and C6, respectively,..." if the binding site is narrow it must be also steric effects. Please discuss that. This is in fact similar to the 1-methyl-Ura.

Response: We have extended the docking analysis and show poses for the relevant nucleobase analogues in the Supplementary Information (Figure S10). There are clear steric effects for some compounds. In others (e.g. 5-fluoro-Ura) the positioning is almost as in the

parent Ura. The discussion in main text was expanded to clarify the role of electronic and steric effects.

d) Table 1: For those unfamiliar with the nucleoside/nucleotide nomenclature it would be very helpful to show the structures too.

Response: We agree with the reviewer. Table 1 was revised as suggested.

REVIEWERS' COMMENTS

Reviewer #1 (Remarks to the Author):

The authors have addressed all concerns in this revised version of their article. I therefore recommend publication in Nature Communications.

Reviewer #2 (Remarks to the Author):

The authors have addressed correctly the questions raised by this referee and the paper can be accepted.

There is one minor thing that the authors should consider. When they are referring to Mannich reaction, the expression "Mannich like condensation" must be Mannich like addition. "Addition" better describes the reaction rather than "condensation"

Revision

The manuscript was revised according to requirements of the Editorial Office and the comments of the reviewers. Changes in the manuscript are highlighted in yellow background. A table/checklist of the Editor's comments including our responses is uploaded in a separate file. Response to the comments of the reviewers are given below in light blue color.

REVIEWERS' COMMENTS

Reviewer #1 (Remarks to the Author):

The authors have addressed all concerns in this revised version of their article. I therefore recommend publication in Nature Communications.

Authors' response: There were no changes needed.

Reviewer #2 (Remarks to the Author):

The authors have addressed correctly the questions raised by this referee and the paper can be accepted.

There is one minor thing that the authors should consider. When they are referring to Mannich reaction, the expression "Mannich like condensation" must be Mannich like addition. "Addition" better describes the reaction rather than "condensation"

Authors' response: The use of "condensation" was avoided and "addition" used instead.